# Cooperative Jamming with AF Relay in Power Monitoring and Communication Systems for Mining

**Wei Meng [1], Yidong Gu [1], Jianjun Bao [1], Li Gan [2], Tao Huang [3] and Zhengmin Kong [2,*]**

1 Tiandi (Changzhou) Automation Co., Ltd., CCTEG Changzhou Research Institute, Changzhou 213015, China
2 School of Electrical Engineering and Automation, Wuhan University, Wuhan 430072, China
3 College of Science and Engineering, James Cook University, Smithfield, QLD 4878, Australia
* Correspondence: zmkong@whu.edu.cn

**Abstract:** In underground mines, physical layer security (PLS) technology is a promising method for the effective and secure communication to monitor the mining process. Therefore, in this paper, we investigate the PLS of an amplify-and-forward relay-aided system in power monitoring and communication systems for mining, with the consideration of multiple eavesdroppers. Explicitly, we propose a PLS scheme of cooperative jamming and precoding for a full-duplex system considering imperfect channel state information. To maximize the secrecy rate of the communications, an effective block coordinate descent algorithm is used to design the precoding and jamming matrix at both the source and the relay. Furthermore, the effectiveness and convergence of the proposed scheme with high channel state information uncertainty have been proven.

**Keywords:** physical layer security; multiple eavesdroppers; full-duplex; underground mine; amplify-and-forward relay

## 1. Introduction

Underground mining promotes the economy's growth, but the dust and poisonous gases formed during mining make it a dangerous and complex operation. Therefore, a reliable communication system is needed to monitor the mining process and communicate with external management offices to ensure the safety and maximum production of the underground mine. Wireless communication technology is applied to realize information exchange in underground mines due to its simple construction.

However, due to the complex structure of underground mines, there exists significant attenuation of radio wave transmission in wireless communications [1]. To solve these problems, relay-aided wireless communications have been studied to improve the reliability and have also been used to enhance the coverage of a broader range of networks. According to the forwarding protocol adopted by the relay, cooperation relay can be divided into amplify-and-forward (AF) and decode-and-forward (DF) relay [2]. AF is the simplest protocol, and it processes the received signals linearly and then forwards them to the destination [3]. Offering a reasonable trade-off between actual implementation costs and benefits, AF is considered the most promising solution [4].

To guarantee the communication rate in wireless communications, full-duplex (FD) relays are studied in Refs. [5,6]. FD technology allows radios to receive and transmit simultaneously on the same frequency band, which can improve spectrum efficiency [7]. Furthermore, in addition to doubling the spectral efficiency of the physical layer, FD can help to solve the throughput losses due to congestion and large point-to-point delays in existing wireless networks.

In addition, due to the openness and sharing of wireless media, any wireless device connected to the communication system can access messages exchanged through the connection, making wireless channels easy to be eavesdropped on and inject with malicious information [8]. Worse still, relay-aided wireless networks may suffer severe security risks

from malicious users since they may eavesdrop on the messages from both the source and the relay. Physical layer security (PLS) can effectively protect the privacy among the transmitter and the legitimate receivers [9]. Shannon conducted pioneering research on secret communications and established the concept of perfect secrecy [10]. Unlike Shannon, Wyner proposed a degraded wiretap channel model in Ref. [11]. After the degraded wiretap channel, the fading wiretap channels and multiple-input-multiple-output wiretap channels have been investigated in Refs. [12,13] and Refs. [14,15], respectively.

The work in Ref. [5] investigated a FD communication system, and the transmission block is divided into an energy harvesting phase and an information transmission phase. Different from Ref. [5], in Ref. [6], an FD is designed to capture energy from the source while forwarding information to the legitimate receivers. With the presence of passive colluding wireless eavesdroppers, Ref. [16] studied the effective secrecy throughput to the physical layer security of in-home and broadband PLC systems. In Ref. [17], the authors investigated the optimal trunk position of FD relay systems with DF and the minimal outage probability criterion considered.

Above all, to the best of our knowledge, the existing contributions fail to ensure secure communications in the challenging FD relay-aided wireless communications scenario in the face of multiple eavesdroppers and imperfect channel state information (CSI). Therefore, in this paper, we propose a PLS scheme of cooperative jamming and precoding for FD-DF relay-assisted wireless communications system considering imperfect CSI, which combines cooperative precoding for legitimate users to improve the quality of legitimate channels and cooperative jamming for illegal users to reduce the quality of eavesdropping channels. Considering the imperfect CSI and multiple eavesdroppers, we use an effective BCD algorithm to design the precoding and jamming matrix at both the source and the relay, in which maximizing the secrecy rate of the FD-AF relay-assisted wireless communications system is emphasized.

This paper is organized as follows. Section 2 describes the system model. The secrecy rate optimization problem is proposed and transformed into a solvable form in Section 3, which also gives the algorithm. Section 4 characterizes the numerical results in different scenarios. Finally, the conclusion is presented in Section 5.

Notation: The $\mathbf{W}^T, \mathbf{W}^H, \text{vec}(\mathbf{W}), \|\mathbf{W}\|$ and $\text{tr}(\mathbf{W})$ denote the transpose, conjugate transpose, vectorization, Frobenius norm, and trace of the matrix, respectively. $\otimes$ denote the Kronecker product, and $\mathbf{W}^K$ represents the $\mathbf{W}\mathbf{W}^H$ along with $\log|\mathbf{E} + \mathbf{CD}| = \log|\mathbf{E} + \mathbf{DC}|$. $\mathbf{E}$ is the identity matrix.

## 2. System Model

Consider a MIMO wireless system, as shown in Figure 1, where a source, a relay, a user, and two eavesdroppers have $N_S$, $N_R$, $N_D$, and $N_E$ channels, respectively. We assume that there is no direct link between the source and the user for the long-distance path loss. For simplicity, the eavesdroppers represent all the eavesdroppers eavesdropping the same legitimate in the same time phase. More specifically, in the first time phase, eavesdroppers eavesdrop $E_1$ message from the source, and in the second time phase, eavesdropper eavesdrop $E_2$ message from the relay.

In wireless communications system, messages are transmitted through MIMO wireless communications channels. We describe each path between two nodes by CSI $\mathbf{H}_{ij,k}$ as the matrix of channel coefficients, where $i \in \{S, R\}$, $j \in \{R, D, E_1, E_2\}$, and $k = 1, 2$ denote the transmitter, receiver, and transmission time phases, respectively. It is worth noting that $\mathbf{H}_{RR,1}$ refers to the self-interference matrix because of self-interference and in the process of transmission $\mathbf{H}_{ij,k}$ stays constant because of the short transmission time.

In this paper, the uncertainty of CSI is taken into consideration, i.e., the CSI of the wireless communications system cannot be perfectly known at the source or the relay due to factors such as the limited capacity of the feedback channel. As a result, the deterministic uncertainty model [18] is introduced to characterize the imperfect CSI, as follows:

$$\mathbf{H}_{ij,k} \in \mathcal{H}_{ij,k} = \left\{ \mathbf{H}_{ij,k} \middle| \mathbf{H}_{ij,k} = \overline{\mathbf{H}}_{ij,k} + \Delta_{ij,k}, \left\| \Delta_{ij,k} \right\| \leq \delta_{ij,k} \right\}, \tag{1}$$

where $\Delta_{ij,k}$ denotes the channel uncertainty as the degree of deviation from the mean CSI $\overline{\mathbf{H}}_{ij,k}$.

In Figure 1, during the first time phase, the source sends confidential signals to the relay while $E_1$ eavesdrops on the signals from the source. To interrupt $E_1$, the relay emits jamming signals to $E_1$. More specifically, the message transmitted by the source is secret data symbol $\mathbf{S} \in \mathcal{CN}(\mathbf{0}, 1)$ precoded by the precoding vector $\mathbf{L} \in \mathbb{C}^{N_S \times 1}$. Then, we can formulate the progress at the source as follows:

$$\mathbf{X}_s = \mathbf{LS}, \tag{2}$$

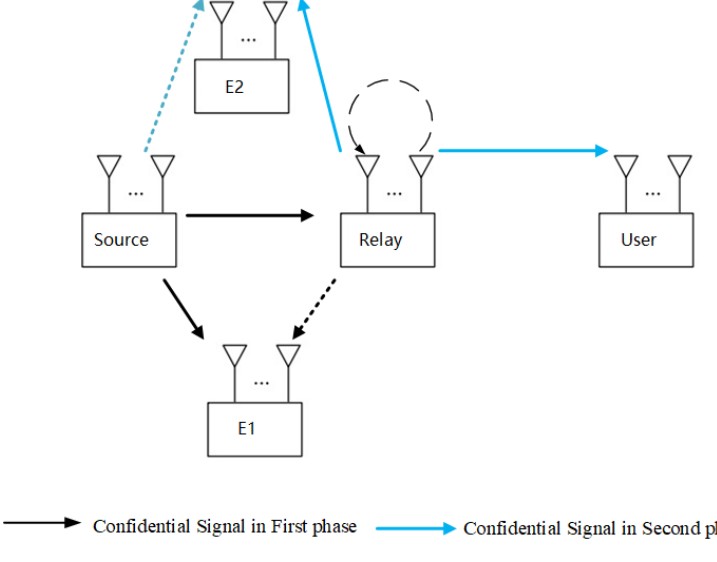

**Figure 1.** Wireless communication system model with a single relay.

Next, we formulate the messages emitted by the relay. Note that the relay in this time phase only emits jamming to disrupt $E_1$ so the messages can be formulated as follows :

$$\mathbf{X}_R = \mathbf{J_1}\mathbf{Z_1}, \tag{3}$$

where we utilize the jamming precoding vector $\mathbf{J_1} \in \mathbb{C}^{N_R \times 1}$ and jamming symbol $\mathbf{Z_1} \in \mathcal{CN}(\mathbf{0}, 1)$.

Considering the self-interference of the relay, we can formulate the messages received by the relay:

$$\mathbf{Y}_{R1} = \mathbf{H}_{SR,1}\mathbf{LS} + \mathbf{H}_I\mathbf{J_1}\mathbf{Z_1} + \mathbf{n}_{R1}, \tag{4}$$

where $\mathbf{n}_{R1}$ is Additive White Gaussian Noise (AWGN) at the relay and $\mathbf{H}_I$ is the self-interference matrix. Meanwhile, $E_1$ eavesdrops on both of the messages from the source and the relay, so the messages eavesdropped by $E_1$ can be expressed as

$$\mathbf{Y}_{E1} = \mathbf{H}_{SE,1}\mathbf{LS} + \mathbf{H}_{RE,1}\mathbf{J_1}\mathbf{Z_1} + \mathbf{n}_{E1}, \tag{5}$$

where $\mathbf{n}_{E1}$ is AWGN at $E_1$.

In the second time phase, the source emits the jamming signals $\mathbf{X}_{S2}$ to $E_2$ where $\mathbf{J_2},\mathbf{Z_2} \in \mathcal{CN}(\mathbf{0},1)$ represent the jamming precoding vector and jamming symbol, respectively.

$$\mathbf{X}_{S2} = \mathbf{J_2}\mathbf{Z_2}, \tag{6}$$

Then, the relay amplifies the messages it received in the first time phase and forwards them to the user,

$$\mathbf{X}_{R2} = \mathbf{G}\mathbf{Y}_{R1} = \mathbf{G}(\mathbf{H}_{SR,1}\mathbf{LS} + \mathbf{H}_I\mathbf{J_1}\mathbf{Z}_1 + \mathbf{n}_{R1}), \tag{7}$$

$$\mathbf{Y}_D = \mathbf{H}_{RD,2}\mathbf{X}_{R2} + \mathbf{n}_D = \mathbf{H}_{RD,2}\mathbf{G}(\mathbf{H}_{SR,1}\mathbf{LS} + \mathbf{H}_I\mathbf{J_1}\mathbf{Z}_1 + \mathbf{n}_{R1}) + \mathbf{n}_D, \tag{8}$$

where $\mathbf{G} \in \mathbb{C}^{N_R \times N_R}$ is the amplifying matrix at the relay and $\mathbf{n}_D$ is AWGN at the users, and $\mathbf{X}_{R2}, \mathbf{Y}_D$ represent the messages transmitted by the relay and received by the users, respectively.

$E_2$ receive both the signals from the relay and the jamming signals from the source, i.e.,

$$\mathbf{Y}_{E2} = \mathbf{H}_{SE,2}\mathbf{J_2}\mathbf{Z_2} + \mathbf{H}_{RE,2}\mathbf{G}(\mathbf{H}_{SR,1}\mathbf{LS} + \mathbf{H}_I\mathbf{J_1}\mathbf{Z}_1 + \mathbf{n}_{R1}) + \mathbf{n}_{E2}, \tag{9}$$

where $\mathbf{n}_{E2}$ is AWGN at $E_2$.

Above all, to formulate the problem in a mathematical form, we calculate the signal-noise ratio (SNR) at the users, $E_1$ and $E_2$, respectively.

$$\mathbf{\Gamma}_D = (\mathbf{H}_{RD,2}\mathbf{G}\mathbf{H}_{SR,1}\mathbf{L})^K \mathbf{Q}_D^{-1}, \tag{10}$$

where $\mathbf{Q}_D = (\mathbf{H}_{RD,2}\mathbf{G}\mathbf{H}_{RR,1}\mathbf{J_1})^K + \sigma_R^2(\mathbf{H}_{RD,2}\mathbf{G})^K + \sigma_D^2\mathbf{E}$.

$$\mathbf{\Gamma}_{E1} = (\mathbf{H}_{SE,1}\mathbf{L})^K \mathbf{Q}_{E1}^{-1}, \tag{11}$$

where $\mathbf{Q}_{E1} = (\mathbf{H}_{RE,1}\mathbf{J_1})^K + \sigma_E^2\mathbf{E}$.

$$\mathbf{\Gamma}_{E2} = (\mathbf{H}_{RE,2}\mathbf{G}\mathbf{H}_{R1}\mathbf{L})^K \mathbf{Q}_{E2}^{-1}, \tag{12}$$

where $\mathbf{Q}_{E2} = (\mathbf{H}_{SE,2}\mathbf{J_2})^K + (\mathbf{H}_{RE,2}\mathbf{G}\mathbf{H}_I\mathbf{J_1})^K + \sigma_R^2(\mathbf{H}_{RE,2}\mathbf{G})^K + \sigma_E^2\mathbf{E}$ and $\sigma_i$ is the noise amplitude of the corresponding AWGN $\mathbf{n}_i$.

Then, we can arrive at the achievable secrecy rate of the legitimate users [11]:

$$R_D = \log|\mathbf{E} + \mathbf{\Gamma}_D|, \tag{13}$$

In the non-colluding strategy, each eavesdropper processes messages individually. Therefore, the achievable secrecy rate of the non-colluding [11] eavesdroppers is

$$R_E = \max\{\log|\mathbf{E} + \mathbf{\Gamma}_{E1}|, \log|\mathbf{E} + \mathbf{\Gamma}_{E2}|\} \tag{14}$$

Finally, we can gain the achievable secrecy rate of the wireless communications system,

$$R_S = R_D - R_E \tag{15}$$

## 3. Optimization Problem Transformation

In this part, the goal is to maximize the secrecy rate of the communication system. Then according to the system model, we can formulate the optimization problem of the secrecy rate with the transmit power constraint as follows.

$$\max_{\mathbf{L},\mathbf{J_1},\mathbf{J_2},\mathbf{G}} \ \min_{\mathbf{H}_{ij,k}\in\mathcal{H}_{ij,k}} \ R_S \tag{16a}$$

$$\text{s.t. } \|\mathbf{L}\|^2 \leqslant P_S, \ \|\mathbf{J_1}\|^2 \leqslant P_S, \ \|\mathbf{J_2}\|^2 \leqslant P_R, \tag{16b}$$

$$tr((\mathbf{GH}_{SR,1}\mathbf{L})^K + (\mathbf{GH}_I\mathbf{J_1})^K + \sigma_R^2\mathbf{G}^K) \leqslant P_R \quad \forall \mathbf{H}_{ij,k} \in \mathcal{H}_{ij,k} \tag{16c}$$

However, due to the non-convexity of the optimization problem, it is difficult to solve. To deal with the high non-convexity of the function $-\log|\cdot|$, the objective function in (16a) is transformed into an equivalent counterpart through the WMMSE algorithm, which can be solved by the BCD method. The following introduces the WMMSE algorithm.

**Lemma 1.** *Define the mean-square error (MSE) matrix*

$$\widehat{\mathbf{N}} \triangleq (\mathbf{TH\text{-}E})^K + \mathbf{TRT}^H \tag{17}$$

*where* $\mathbf{R} \succ \mathbf{0}$. *Then we have*

$$-\log|\mathbf{N}| = \max_{\mathbf{K}\succ 0} \ \log|\mathbf{K}| - tr(\mathbf{KN}) + tr(\mathbf{E}) \tag{18}$$

$$\log\left|\mathbf{I} + \mathbf{R}^{-1}\mathbf{H}^K\right| = \max_{\mathbf{K}\succ 0,\mathbf{T}} \ \log|\mathbf{K}| - tr(\mathbf{K}\widehat{\mathbf{N}}) + tr(\mathbf{E}) \tag{19}$$

To reformulate the parts of $-\log|\cdot|$ in the objective function, we apply Lemma 1 on (13) and introduce the MSE matrix $\mathbf{N}_i$ and auxiliary matrices $\mathbf{K}_i$, $\mathbf{T}_i$, which have been defined in (17) and (19). So, the achievable secrecy rate of the legitimate can be reorganized as

$$R_D = \log|\mathbf{E} + \mathbf{\Gamma}_D| = \log\left|\mathbf{E} + (\mathbf{H}_{RD,2}\mathbf{GH}_{SR,1}\mathbf{L})^K\mathbf{Q}_D^{-1}\right|$$

$$= \max_{\mathbf{K}_D\succ 0,\mathbf{D}_D} \ \log|\mathbf{K}_D| - tr(\mathbf{K}_D\mathbf{N}_D) + tr(\mathbf{E}) \tag{20}$$

where

$$\mathbf{N}_D = (\mathbf{T}_D\mathbf{H}_{RD,2}\mathbf{GH}_{SR,1}\mathbf{L} - \mathbf{E})^K + \mathbf{T}_D\mathbf{Q}_D\mathbf{T}_D^H \tag{21}$$

Applying Lemma 1 on (14), the achievable rates of $E_1$ and $E_2$ can be transformed as (22) and (23).

$$-\log|\mathbf{E} + \mathbf{\Gamma}_1| = \log|\mathbf{Q}_{E1}| - \log\left|(\mathbf{H}_{SE,1}\mathbf{L})^K + \mathbf{Q}_{E1}\right|$$

$$= \underbrace{\log\left|\mathbf{E} + \sigma_E^{-2}(\mathbf{H}_{RE,1}\mathbf{J_1})^K\right|}_{C_{E11}} - \underbrace{\log\left|\mathbf{E} + \sigma_E^{-2}\left((\mathbf{H}_{SE,1}\mathbf{L})^K + (\mathbf{H}_{RE,1}\mathbf{J_1})^K\right)\right|}_{C_{E12}} \tag{22}$$

$$
\begin{aligned}
-\log|\mathbf{E}+\boldsymbol{\Gamma}_2| &= \log|\mathbf{Q}_{E2}| - \log\left|(\mathbf{H}_{RE,2}\mathbf{GH}_I\mathbf{L})^K + \mathbf{Q}_{E2}\right| \\
&= \underbrace{\log\left|\mathbf{E}+\sigma_E^{-2}\left((\mathbf{H}_{SE,2}\mathbf{J_2})^K + (\mathbf{H}_{RE,2}\mathbf{GH}_I\mathbf{J_1})^K + \sigma_R^2(\mathbf{H}_{RE,2}\mathbf{G})^K\right)\right|}_{C_{E21}} + \\
&\quad \underbrace{-\log\left|\mathbf{E}+\sigma_E^{-2}\left((\mathbf{H}_{RE,2}\mathbf{GH}_I\mathbf{L})^K + (\mathbf{H}_{SE,2}\mathbf{J_2})^K + (\mathbf{H}_{RE,2}\mathbf{GH}_I\mathbf{J_1})^K + \sigma_R^2(\mathbf{H}_{RE,2}\mathbf{G})^K\right)\right|}_{C_{E22}}
\end{aligned}
\tag{23}
$$

Then, the auxiliary variables $C_{E11}$, $C_{E12}$, $C_{E21}$ and $C_{E22}$ can be rewritten according to Lemma 1 as

$$
C_{E11} = \max_{\mathbf{K}_{E11}\succ 0,\mathbf{T}_{E1}} \log|\mathbf{K}_{E11}| - \mathrm{tr}(\mathbf{K}_{E11}\mathbf{N}_{E11}) + \mathrm{tr}(\mathbf{E})
\tag{24}
$$

$$
C_{E12} = \max_{\mathbf{K}_{E12}\succ 0} \log|\mathbf{K}_{E12}| - \mathrm{tr}(\mathbf{K}_{E12}\mathbf{N}_{E12}) + \mathrm{tr}(\mathbf{E})
\tag{25}
$$

$$
C_{E21} = \max_{\mathbf{K}_{E21}\succ 0,\mathbf{T}_{E2}} \log|\mathbf{K}_{E21}| - \mathrm{tr}(\mathbf{K}_{E21}\mathbf{N}_{E21}) + \mathrm{tr}(\mathbf{E})
\tag{26}
$$

$$
C_{E22} = \max_{\mathbf{K}_{E22}\succ 0} \log|\mathbf{K}_{E22}| - \mathrm{tr}(\mathbf{K}_{E22}\mathbf{N}_{E22}) + \mathrm{tr}(\mathbf{E})
\tag{27}
$$

where

$$
\mathbf{N}_{E11} = (\mathbf{DT}_{E1}\mathbf{H}_{RE,1}\mathbf{J_1} - \mathbf{E})^K + \sigma_E^2\mathbf{T}_{E1}^K
$$

$$
\mathbf{N}_{E12} = \sigma_E^{-2}\left((\mathbf{H}_{SE,1}\mathbf{L})^K + (\mathbf{H}_{RE,1}\mathbf{J_1})^K\right) + \mathbf{E}
$$

$$
\mathbf{N}_{E21} = (\mathbf{T}_{E21}\mathbf{H}_{SE,2}\mathbf{J_2}\mathbf{X} + \mathbf{T}_{E22}\mathbf{H}_{RE,2}\mathbf{GH}_I\mathbf{VX} +
$$
$$
\sigma_R\mathbf{T}_{E23}\mathbf{H}_{RE,2}\mathbf{G} - \mathbf{E})^K + \sigma_E^2\left(\mathbf{T}_{E21}^K + \mathbf{T}_{E22}^K + \mathbf{T}_{E23}^K\right)
$$

$$
\mathbf{N}_{E22} = \sigma_E^{-2}((\mathbf{H}_{RE,2}\mathbf{GH}_I\mathbf{L})^K + (\mathbf{H}_{SE,2}\mathbf{J_2})^K + (\mathbf{H}_{RE,2}\mathbf{GH}_I\mathbf{J_1})^K + \sigma_R^2(\mathbf{H}_{RE,2}\mathbf{G})^K) + \mathbf{E}
$$

and note the decomposition $\mathbf{T}_{E2} = \begin{bmatrix} \mathbf{T}_{E21} & \mathbf{T}_{E22} & \mathbf{T}_{E23} \end{bmatrix}$ and $\mathbf{X} = \begin{bmatrix} 1 & \mathbf{0} \end{bmatrix} \in \mathbb{C}^{1\times Nr}$.

After substituting (24)–(27) into (16a), the secrecy rate of the system is equivalently rewritten as

$$
\max_{\mathbf{L},\mathbf{J_1},\mathbf{J_2},\mathbf{G},\mathbf{K}_i\succ 0,\mathbf{T}_i} \min_{\mathbf{H}_{ij,k}\in\mathcal{H}_{ij,k}} f(\mathbf{L},\mathbf{J_1},\mathbf{J_2},\mathbf{G},\mathbf{S_i},\mathbf{D_i})
\tag{28}
$$

$$
\text{s.t. (16c)}
\tag{29}
$$

$$
\begin{aligned}
f &\triangleq \log|\mathbf{K}_D| - \mathrm{tr}(\mathbf{K}_D\mathbf{N}_D) + \min\{\log|\mathbf{K}_{E11}| - \mathrm{tr}(\mathbf{K}_{E11}\mathbf{N}_{E11}) + \log|\mathbf{K}_{E12}| - \mathrm{tr}(\mathbf{K}_{E12}\mathbf{N}_{E12}), \\
&\quad \log|\mathbf{K}_{E21}| - \mathrm{tr}(\mathbf{K}_{E21}\mathbf{N}_{E21}) + \log|\mathbf{K}_{E22}| - \mathrm{tr}(\mathbf{K}_{E22}\mathbf{N}_{E22})\}
\end{aligned}
\tag{30}
$$

where the function $f(\mathbf{L},\mathbf{J_1},\mathbf{J_2},\mathbf{G},\mathbf{K_i},\mathbf{T_i})$ is defined in (30).

To solve the proposed problem and constraint (16c), the slack variables $\beta_i$ ($i \in \{T, E11, E12, E21, E22, P\}$) are introduced to transform (28) into an optimization problem.

$$\mathrm{tr}(\mathbf{K}_i \mathbf{N}_i) \leq \beta_i, \forall \mathbf{H}_{ij,k} \in \mathcal{H}_{ij,k} \tag{31}$$

We can further rewrite the problem (28) as

$$\max_{\mathbf{L},\mathbf{J_1},\mathbf{J_2},\mathbf{G},\mathbf{K}_i \succ 0, \mathbf{T}_i} g(\mathbf{L},\mathbf{J_1},\mathbf{J_2},\mathbf{G},\mathbf{S_i},\mathbf{D_i}) \tag{32}$$

$$\text{s.t. (16c), (31)} \tag{33}$$

$$g \triangleq \log|\mathbf{K}_D| - \beta_D + \min\{\log|\mathbf{K}_{E11}| - \beta_{E11} + \log|\mathbf{K}_{E12}| - \beta_{E12},$$
$$\log|\mathbf{K}_{E21}| - \beta_{E21} + \log|\mathbf{K}_{E22}| - \beta_{E22}\} \tag{34}$$

where $g(\mathbf{L},\mathbf{J_1},\mathbf{J_2},\mathbf{G},\mathbf{K_i},\mathbf{T_i})$ is defined in (34), respectively. However, the semi-infinite inequalities (31) are non-convex and need further transformation. In the next step, (31) is transformed into a convex form. In fact, all the inequalities $\mathrm{tr}(\mathbf{K}_i \mathbf{N}_i) \leq \beta_i$ can be transformed into a convex form in a similar way. Such as, when $i = D$, the semi-definite constraint $\mathrm{tr}(\mathbf{K}_D \mathbf{N}_D)$ can be rewritten as

$$\mathrm{tr}(\mathbf{K}_D \mathbf{N}_D) = \left\| \underbrace{\begin{bmatrix} \mathrm{vec}(\mathbf{F}_D(\mathbf{T}_D \mathbf{H}_{RD,2}\mathbf{GH}_{SR,1}\mathbf{L} - \mathbf{E})) \\ \mathrm{vec}(\mathbf{F}_D \mathbf{T}_D \mathbf{H}_{RD,2}\mathbf{GH}_{RR,1}\mathbf{J_1}) \\ \mathrm{vec}(\sigma_R \mathbf{F}_D \mathbf{T}_D \mathbf{H}_{RD,2}\mathbf{G}) \\ \mathrm{vec}(\sigma_D \mathbf{F}_D \mathbf{T}_D) \end{bmatrix}}_{\phi_D} \right\|^2 \tag{35}$$

by applying $\mathbf{T}_D = \mathbf{F}_D^H \mathbf{F}_D$ and the equality $\mathrm{tr}(\mathbf{W}^K) = \|\mathrm{vec}(\mathbf{W})\|^2$.

Then we need to extract the uncertain CSI from (35).

$$\phi_D = \bar{\phi}_D + \underbrace{\sum_j \mathbf{\Omega}_{Dj}\mathrm{vec}(\mathbf{\Delta}_j)}_{\mathbf{\Delta}_D} + \underbrace{\sum_k \alpha_k \mathrm{vec}(\mathbf{\Delta}_{k1})\mathrm{vec}^H(\mathbf{\Delta}_{k2})}_{\widetilde{\mathbf{\Delta}}_D} \tag{36}$$

where the identity $\mathrm{vec}(\mathbf{ABC}) = (\mathbf{C}^T \otimes \mathbf{A})\mathrm{vec}(\mathbf{B})$ is applied and $j \in \{RR,1; RD,2; SR,1\}$. Note that $k1, k2$ denote the coupling parts of CSI in $\phi_D$ in the $\widetilde{\mathbf{\Delta}}_D$ part. In fact, the uncertainty of the CSI is small enough to make its quadratic forms negligible. As a result, the $\phi_D$ can be represented as its asymptotic form as

$$\phi_D = \bar{\phi}_D + \underbrace{\sum_j \mathbf{\Omega}_{Dj}\mathrm{vec}(\mathbf{\Delta}_j)}_{\mathbf{\Delta}_D} \tag{37}$$

where

$$\bar{\phi}_D = \begin{bmatrix} \mathrm{vec}(\mathbf{F}_D(\mathbf{T}_D \bar{\mathbf{H}}_{RD,2}\mathbf{G}\bar{\mathbf{H}}_{SR,1}\mathbf{L} - \mathbf{E})) \\ \mathrm{vec}(\mathbf{F}_D \mathbf{T}_D \bar{\mathbf{H}}_{RD,2}\mathbf{G}\bar{\mathbf{H}}_I \mathbf{J_1}) \\ \mathrm{vec}(\sigma_R \mathbf{F}_D \mathbf{T}_D \bar{\mathbf{H}}_{RD,2}\mathbf{G}) \\ \mathrm{vec}(\sigma_D \mathbf{F}_D \mathbf{T}_D) \end{bmatrix} \tag{38}$$

$$\boldsymbol{\Omega}_{DSR,1} = \begin{bmatrix} \mathbf{L}^T \otimes \mathbf{F}_D \mathbf{T}_D \bar{\mathbf{H}}_{RD,2} \mathbf{G} \\ \mathbf{0} \\ \mathbf{0} \\ \mathbf{0} \end{bmatrix} \tag{39}$$

$$\boldsymbol{\Omega}_{DSD,2} = \begin{bmatrix} \left(\mathbf{G}\bar{\mathbf{H}}_{SR,1}\mathbf{L}\right)^T \otimes \mathbf{F}_D \mathbf{T}_D \\ \left(\mathbf{G}\bar{\mathbf{H}}_{RR,1}\mathbf{J}_1\right)^T \otimes \mathbf{F}_D \mathbf{T}_D \\ \sigma_R \mathbf{G}^T \otimes \mathbf{F}_D \mathbf{T}_D \\ \mathbf{0} \end{bmatrix} \tag{40}$$

$$\boldsymbol{\Omega}_{DRR,1} = \begin{bmatrix} \mathbf{0} \\ \mathbf{J_1}^T \otimes \mathbf{F}_D \mathbf{D}_D \bar{\mathbf{H}}_{RD,2} \mathbf{G} \\ \mathbf{0} \\ \mathbf{0} \end{bmatrix} \tag{41}$$

Then, we exploit the Schur complement lemma to recast the constraint (31) as a matrix inequality by substituting (35) and (37).

$$\begin{bmatrix} \beta_D & \bar{\phi}_D^H \\ \bar{\phi}_D & \mathbf{E} \end{bmatrix} \succ - \begin{bmatrix} 0 & \boldsymbol{\Delta}_D^H \\ \boldsymbol{\Delta}_D & 0 \end{bmatrix} \tag{42}$$

To eliminate the $\boldsymbol{\Delta}_D$, the sign-definiteness lemma is applied.

**Lemma 2.** *Defined matrix* $\mathbf{U}$ *and* $\{\mathbf{P}_i, \mathbf{Q}_i\}$, $i \in \{1, 2, \ldots, N\}$ *with* $\mathbf{U} = \mathbf{U}^H$, *the semi-infinite Linear Matrix Inequality (LMI) of the form*

$$\mathbf{U} \succ \sum_{i}^{N} \left( \mathbf{P}_i^H \mathbf{Y}_i \mathbf{Q}_i + \mathbf{Q}_i^H \mathbf{Y}_i^H \mathbf{P}_i \right), \ \|\mathbf{Y}_i\| \leq \delta_i \tag{43}$$

Holds if and only if there exist nonnegative real numbers $\lambda_1, \lambda_2, \ldots, \lambda_N$ such that

$$\begin{bmatrix} \mathbf{U} - \sum_{i=1}^{N} \lambda_i \mathbf{Q}_i^H \mathbf{Q}_i & -\delta_1 \mathbf{P}_1^H & \cdots & -\delta_N \mathbf{P}_N^H \\ -\delta_1 \mathbf{P}_1 & \delta_1 \mathbf{E} & \cdots & \mathbf{0} \\ \vdots & \vdots & \ddots & \vdots \\ -\delta_N \mathbf{P}_N & \mathbf{0} & \cdots & \delta_N \mathbf{E} \end{bmatrix} \succ \mathbf{0} \tag{44}$$

Appropriately choose the parameters below

$$\mathbf{U}_D = \begin{bmatrix} \beta_D & \bar{\phi}_D^H \\ \bar{\phi}_D & \mathbf{I} \end{bmatrix} \tag{45}$$

$$\mathbf{Q}_{D1} = \mathbf{Q}_{D2} = \mathbf{Q}_{D3} = [-10] \tag{46}$$

$$\mathbf{P}_{D1} = \begin{bmatrix} \mathbf{0} & \boldsymbol{\Omega}_{DSR,1}^H \end{bmatrix} \tag{47}$$

$$\mathbf{P}_{D2} = \begin{bmatrix} \mathbf{0} & \boldsymbol{\Omega}_{DSD,2}^H \end{bmatrix} \tag{48}$$

$$\mathbf{P}_{D3} = \begin{bmatrix} \mathbf{0} & \boldsymbol{\Omega}_{DRR,1}^H \end{bmatrix} \tag{49}$$

Apply Lemma 2 to transform (42) as

$$
\left[\begin{array}{cc} \left[\begin{array}{cc} \beta_D-\lambda_{D1}-\lambda_{D2}-\lambda_{D3} & \bar{\phi}_D^H \\ \bar{\phi}_D & \mathbf{E} \end{array}\right] & \mathbf{\Theta}_D^H \\ \mathbf{\Theta}_D & \mathrm{diag}(\lambda_{D1}\mathbf{E},\lambda_{D2}\mathbf{I},\lambda_{D3}\mathbf{I}) \end{array}\right] \succ 0 \tag{50}
$$

where $\mathbf{\Theta}_D = -[\delta_{DSR,1}\mathbf{P}_{D1}^T, \delta_{DSD,2}\mathbf{P}_{D2}^T, \delta_{DRR,1}\mathbf{P}_{D3'}^T]^T$. Similarly, the other constraint $\mathrm{tr}(\mathbf{K}_i\mathbf{N}_i) \leq \beta_i$ is written as follows.

$$
\left[\begin{array}{cc} \left[\begin{array}{cc} \beta_i - \sum\limits_{k=l}^{j} \lambda_k & \bar{\phi}_i^H \\ \bar{\phi}_i & \mathbf{E} \end{array}\right] & \mathbf{\Theta}_i^H \\ \mathbf{\Theta}_i & [\lambda_l\mathbf{E}, \cdots, \lambda_j\mathbf{E}] \end{array}\right] \succ 0 \tag{51}
$$

By assembling all the components, the problem can now be written as

$$
\max_{\mathbf{L},\mathbf{J_1},\mathbf{J_2},\mathbf{G},\mathbf{F}_i\succ 0,\mathbf{T_i},\lambda_i\geq 0,\beta_i} h(\mathbf{L},\mathbf{J_1},\mathbf{J_2},\mathbf{G},\mathbf{F_i},\mathbf{T_i},\lambda_i,\beta_i) \tag{52}
$$

$$
\text{s.t. } (16c),\ (50),\ (51) \tag{53}
$$

$$
h \overset{\Delta}{=} 2\log|\mathbf{F}_D| - \beta_D + \min\{2\log|\mathbf{F}_{E11}| - \beta_{E11} + 2\log|\mathbf{F}_{E12}| - \beta_{E12},\\ 2\log|\mathbf{F}_{E21}| - \beta_{E21} + 2\log|\mathbf{F}_{E22}| - \beta_{E22}) \tag{54}
$$

where the function $h(\mathbf{L},\mathbf{J_1},\mathbf{J_2},\mathbf{G},\mathbf{F_i},\mathbf{T_i},\lambda_i,\beta_i)$ is defined in (54). The problem (52) remains non-convex. However, it becomes a convex optimization problem when fixing some of the optimization variables. In other words, after proper manipulations, its sub-problems can become convex, which are readily solvable. Therefore, a BCD algorithm is employed to solve the nonconvex problem (52), which is summarized in Algorithm 1.

---

**Algorithm 1** AN-BF scheme to solve the optimization problem

---

**input** $l= 0$, precoding vector $\mathbf{L}=\mathbf{L}^{(0)}$; jamming precoding vector $\mathbf{J_1}=\mathbf{J_1}^{(0)},\mathbf{J_2}=\mathbf{J_2}^{(0)}$; $\mathbf{F}_i = \mathbf{F}_i^{(0)}, \mathbf{G} = \mathbf{G}^{(0)}$;

**repeat**

　1: Begin BCD to deal with the (52) with $\mathbf{L}=\mathbf{L}^{(l-1)},\mathbf{J_1}=\mathbf{J_1}^{(l-1)},\mathbf{J_2}=\mathbf{J_2}^{(l-1)}$; $\mathbf{F}_i = \mathbf{F}_i^{(l-1)}, \mathbf{G} = \mathbf{G}^{(l-1)}$, and gain the $\mathbf{D}_i^{(l)}$;

　2: Solve (52) with $\mathbf{L}=\mathbf{L}^{(l-1)},\mathbf{J_1}=\mathbf{J_1}^{(l-1)},\mathbf{J_2}=\mathbf{J_2}^{(l-1)}$; $\mathbf{D}_i = \mathbf{D}_i^{(l)}, \mathbf{G} = \mathbf{G}^{(l-1)}$, and gain the $\mathbf{F}_i^{(l)}$;

　3: Solve (52) to attain $\mathbf{J_1}^{(l)},\mathbf{J_2}^{(l)}$ and $\mathbf{L}^{(l)}$ with $\mathbf{D}_i = \mathbf{D}_i^{(l)}, \mathbf{G} = \mathbf{G}^{(l-1)}, \mathbf{F}_i = \mathbf{F}_i^{(l)}$;

　4: Solve (52) to gain $\mathbf{G}^{(l)}$ with $\mathbf{L}=\mathbf{L}^{(l)},\mathbf{J_1}=\mathbf{J_1}^{(l)},\mathbf{J_2}=\mathbf{J_2}^{(l)}, \mathbf{F}_i = \mathbf{F}_i^{(l)}, \mathbf{D} = \mathbf{D}^{(l)}$;

**until** $\left|y^{(l)} - y^{(l-1)}\right| \leq \varepsilon$.

---

## 4. Results

In this section, numerical simulations are provided to evaluate the performance of the proposed scheme in terms of the average secrecy rate. In this part, we consider a wireless communications system with $N_S = N_R = N_D = N_E = N = 2$. Besides, for simplicity, the CSI uncertainty bound $\delta_{ij,k}$ is represented as the corresponding determinant of mean CSI multiplied by one certain coefficient, or $\delta_{ij,k} = \mu\left\|\overline{\mathbf{H}}_{ij,k}\right\|$.

Figure 2 portrays the average secrecy rate versus numbers of iterations with $P_S = P_R = P = 10$ dB. By the proposed scheme, the average secrecy rate always converges within about 40 iterations. It indicates that the CSI uncertainty has a destructive effect on the

secrecy rate and the BCD algorithm converges faster with larger uncertainty. Additionally, the proposed scheme achieves a better average secrecy rate with more ports of legitimate users and fewer ports of eavesdroppers, which is especially obvious in small uncertainty scenarios. It can be explained that the number of ports suggests the ability to receive or intercept the information.

Figure 3 shows the impact of a different transmit power of the proposed scheme. It can be observed that the average secrecy rate increases with the increase of transmitting power. In addition, it is observed that the security rate does not improve significantly when the transmitted power is more than 10 dB under the condition of more ports of eavesdroppers and greater CSI uncertainty. It can be explained that the increase in transmitting power increases the capacity of not only legitimate users but also eavesdroppers, resulting in a slight change in the security rate.

We compare the proposed schemes with a similar one without jamming by presenting the numerical results in Figure 4. Our proposed scheme achieves better performance in terms of the average secrecy rate, especially with lower uncertainty and higher transmit power. Therefore, to some extent, jamming can disturb the interception of eavesdroppers even with higher uncertainty.

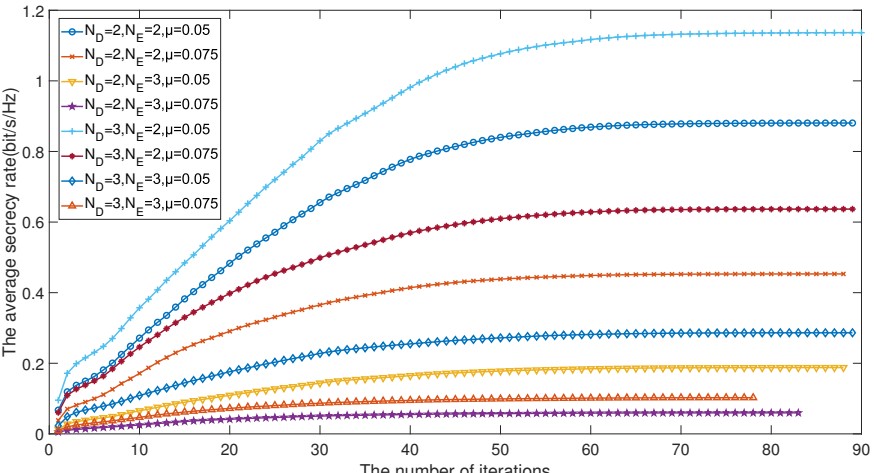

**Figure 2.** Average secrecy rate versus the number of iterations, a comparison of different ports number and CSI uncertainty.

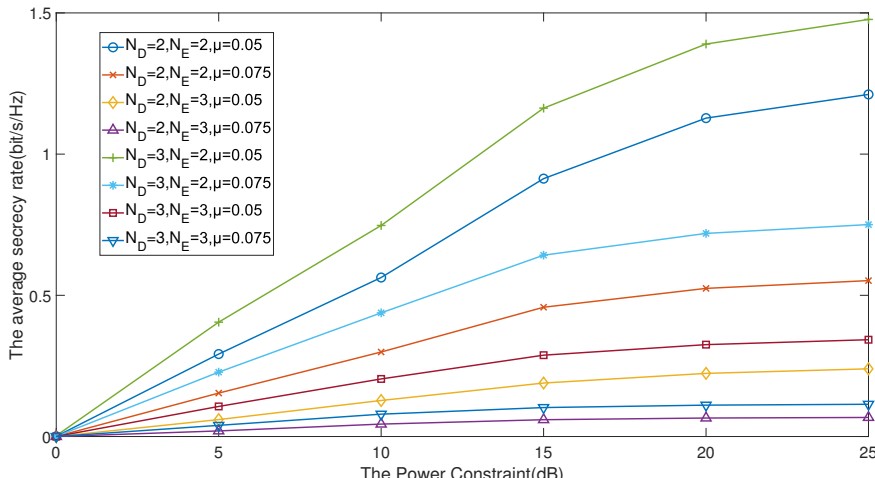

**Figure 3.** Average secrecy rate versus power constraint, a comparison of different antenna numbers and CSI uncertainty.

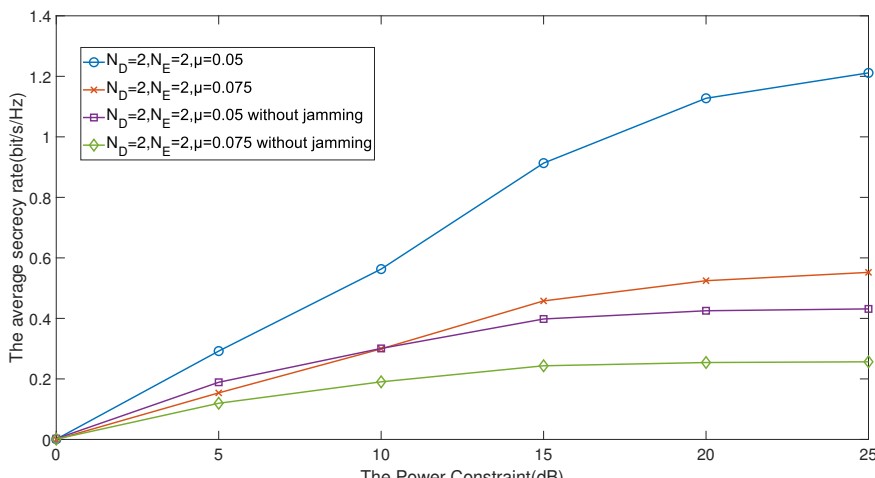

**Figure 4.** Average secrecy rate versus power constraint comparison of different schemes.

## 5. Discussion

In this paper, the precoding jamming scheme has been proposed to enhance the security of AF relay-aided power monitoring and communication systems, where the CSI uncertainty and colluding eavesdroppers are considered. Such a system can be used in an underground mining process to guarantee the communication with management offices to ensure the safety. The scheme combined cooperative precoding for users and cooperative jamming for eavesdroppers. Numerical results have shown that the proposed scheme outperforms the scheme without jamming. Furthermore, the effectiveness of the proposed scheme with high CSI uncertainty has been proven.

**Author Contributions:** Methodology, W.M.; writing—original draft, Y.G. and L.G.; formal analysis, J.B.; validation, T.H. and Z.K.; writing—review and editing, T.H. and Z.K. All authors have read and agreed to the published version of the manuscript.

**Funding:** This research was supported by the National Natural Science Foundation of China under Grant 62173256.

**Data Availability Statement:** Not applicable.

**Conflicts of Interest:** The authors declare no conflict of interest.

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
