# Peer review of "Cooperative Jamming with AF Relay in Power Monitoring and Communication Systems for Mining"

_electronics, doi:10.3390/electronics12041057_

Round 1

Reviewer 1 Report

The authors present physical layer security of amplify and forward relay-aided systems in power monitoring, communication system, and mining. The authors have supported their study by providing a detailed MIMO system model and the simulation results.

Authors should mention the full form of the acronym at first use in the text. There is no full form for CSI at line number 9.

There is no need to introduce an abbreviation for wireless communication i.e. WLC

The authors have mentioned CTF uncertainty. Do they mean CSI uncertainty? It is suggested to use only one form of information.

Authors have also shown the impact of different transmit power levels. It can be observed that the average secrecy rate increases with the increase of transmitting power... 

.

Reviewer 2 Report

The work is very interesting, but some definitions are missing, it is not possible to follows from equation to equation, some work must be done in that. Some variables just appears without the proper definition, it is probably clear for the authors but almost impossible for the reader. It can be correct but there is information missing for a reader. After a certain point I could not follow the development.

Round 2

Reviewer 2 Report

The comments from the first review report are properly addressed.